# Security Performance Analysis of LEO Satellite Constellation Networks under DDoS Attack

**DOI:** 10.3390/s22197286

**Published:** 2022-09-26

**Authors:** Yan Zhang, Yong Wang, Yihua Hu, Zhi Lin, Yadi Zhai, Lei Wang, Qingsong Zhao, Kang Wen, Linshuang Kang

**Affiliations:** 1College of Electronic Engineering, National University of Defense Technology, Hefei 230037, China; 2Anhui Province Key Laboratory of Electronic Restriction, Hefei 230037, China

**Keywords:** LSCNs, DDoS, space-time graph model, security performance, anti-DDoS

## Abstract

Low Earth orbit satellite constellation networks (LSCNs) have attracted significant attention around the world due to their great advantages of low latency and wide coverage, but they also bring new challenges to network security. Distributed denial of service (DDoS) attacks are considered one of the most threatening attack methods in the field of Internet security. In this paper, a space-time graph model is built to identify the key nodes in LSCNs, and a DDoS attack is adopted as the main means to attack the key nodes. The scenarios of two-satellite-key-node and multi-satellite-key-node attacks are considered, and their security performance against DDoS attacks is also analyzed. The simulation results show that the transmission path of key satellite nodes will change rapidly after being attacked, and the average end-to-end delay and packet loss are linearly related to the number of key-node attacks. This work provides a comprehensive analysis of the security performance of LSCNs under a DDoS attack and theoretical support for future research on anti-DDoS attack strategies for LSCNs.

## 1. Introduction

Low Earth orbit (LEO) satellite constellation networks have attracted people’s attention due to their benefits of low latency and extensive coverage [1,2,3,4,5,6,7,8,9]. Countries around the world have proposed LEO constellation plans, showing their ambition to join the “space race” of satellite networks. A number of studies about satellite communication systems have begun both domestically and internationally as a result of the significant advancement in LEO satellite communication technology. LEO satellite constellations such as OneWeb, Kupier, and Starlink are the most prevalent [10,11,12]. The most developed low Earth orbit satellite constellation network (LSCN) system at the moment is SpaceX’s Starlink satellite network.

Compared to terrestrial networks, LSCNs have periodicity and regularity. However, although these characteristics bring convenience to the research of LSCNs, they also make LSCNs extremely vulnerable to several types of threats and attacks [13]. For instance, an adversary can take advantage of the LSCN’s global coverage to turn its advantages into vulnerabilities and manipulate ground users to form botnets to launch DDoS attacks from multiple locations. This kind of attack threat has been regularly exploited in terrestrial networks and has caused significant damage, and it is quite likely to be used to attack LSCNs in the future.

Due to frequent link switching, the network topology also changes accordingly. How to obtain the information of key network nodes has become one of the important research directions of many scholars in satellite network security. Wei et al. [14] conducted an evaluation based near the center of the rapid assessment of the importance of satellite network nodes. The study indicated that the results of this method of assessment are more reasonable. To improve the survivability of LSCNs, Wang et al. [15] utilized time-cumulative graph techniques (C-TVG) to compute the betweenness centrality of each node in the graph in the modeling network but only considered betweenness centrality. Considering the differences in inter-layer connectivity relationships, Xu et al. [16] proposed the dynamic supra-adjacency matrix (DSAM) temporal network model to measure the importance of satellite nodes. Finally, experimental simulations of the Iridium and Orbcomm constellations demonstrated that the DSAM method has a relatively accurate recognition rate and high stability.

At present, there are relatively few studies on satellite network security, especially the DDoS attack [17,18,19,20]. The latest research progress on LSCN DDoS attacks was the first volumetric DDoS attack against next-generation LSCNs: ICARUS [21]. The Starlink constellation was simulated to study single-link and multi-link DDoS attacks. Single-link attacks were divided into attacking uplinks, downlinks, and inter-satellite links; multi-link attacks were based on the calculation and analysis of all links of the satellite Internet between two regions and attacks on the bottleneck link that connect the two regions so as to achieve regional communication blocking. Domestic research on satellite network security mainly focuses on DDoS protection and detection. Aiming at protecting satellite Internet from DDoS attacks, Guo et al. [22] proposed a blockchain-based distributed collaborative entrance defense (DCED) framework, with which network traffic characteristics can be recorded and aggregated at the entrances of satellite Internet. The results show that the framework can accurately identify attack traffic in 1500 ms, and the framework is more effective than other similar DDoS methods.

However, there is little research on the security performance of LSCNs at present, which greatly limits the comprehensive development of LSCNs. Therefore, the space-time graph model is firstly used to determine the key nodes in the satellite network, a DDoS attack is used as the main means to attack the key nodes in the link, and the security performance of the entire satellite network against DDoS attacks is analyzed. Finally, the co-simulation in the scenario is confirmed. It has certain reference value for analyzing the security of LSCNs.

In the following sections, we first introduce related works on satellite key-node identification and satellite network DDoS attacks. Section 2 describes the methods used. Next, the simulated scenario and the parameter settings are given in Section 3. Section 4 analyzes the security performance of LSCNs after being attacked. Finally, we provide the conclusions.

## 2. Materials and Methods

### 2.1. Space-Time Graph Model

Studying the identification of key nodes in a satellite network is the basis of network vulnerability analysis, and it is also an important step in attacking LSCNs with DDoS. As a spatial information network, LSCNs are dynamic and connected, and traditional static graph models cannot model their highly dynamic network topology. Therefore, the concept of topology snapshot was formally proposed. Suppose there are N satellite nodes in a satellite network, $V=\left\{v_{1},v_{2},..,v_{N}\right\}$ represents the set of all satellite nodes. Divide the satellite network into time slots {1,2,…,T} with the same interval in a period; then, in the time slot t, the network topology of the satellite can be expressed as $\;G^{t}=\left\{V^{t},E^{t}\right\}$. In this time slot, $V^{t}$ represents the set of all satellite nodes, and $E^{t}$ represents the set of all links. So the set of satellite network topology snapshot sequences can be represented by $\left\{G^{1},G^{2},\ldots ,G^{T}\right\}$. However, there is no end-to-end path between some node pairs in the snapshot, and the network topology composed of topological snapshot sequences is not connected.

In order to solve the connectivity problem, we transform the satellite network topology snapshot sequence $\left\{G^{1},G^{2},\ldots ,G^{T}\right\}$ into a space-time graph model G=(V,ℰ)  [23,24]. In the space-time graph model, the satellite network is divided into T+1 layers, the satellite nodes of each layer are $V=\left\{v_{1},v_{2},..,v_{N}\right\}$, and so the entire space-time graph contains N×(T+1) nodes. Each adjacent layer can be connected by a space link and a time link. If a directed link $\vec{v_{i}^{t}v_{j}^{t}}\;\in E^{t}$ exists, a space link $\vec{v_{i}^{t-1}v_{j}^{t}}\;\wedge \;\left(i\neq j\right)$ will be added between the adjacent layers, indicating that the node $v_{i}$ can send data to $v_{j}$ during time period [t−1,t). The red path represents the space-time path for data sent from node $v_{2}$ to reach node $v_{5}$ through 4 time slots. Compared with other models, the space-time graph model can analyze the topology of a satellite dynamic network from the time dimension and space dimension and identify the key nodes in the constructed network. The space-time graph model is shown in Figure 1.

### 2.2. Distributed Denial of Service (DDoS) Attacks

Denial of service (DoS) is an attack method that employs network protocol weaknesses to consume target resources and prevent victim hosts or network services from operating normally. However, because of the assault’s sluggish tempo and restricted attack area, the adversary is unable to initiate a large-scale flood. DDoS is a scaled-up version of a DoS attack. The principle is that hackers hijack a large number of normal hosts to make them puppet hosts and send flood packets to the server to consume network resources and link bandwidth. The DDoS assault is currently regarded as the most dangerous attack method in the area of Internet security. The adversary can use the asymmetry of resources to generate large amounts of attack traffic, which eventually leads to a server crash. Figure 2 illustrates the DDoS attacks on LSCNs. The specific manifestations are as follows:Generate a lot of useless data, blocking satellite communication and making the attacked host unable to respond to user requests normally;Utilize the flaws in the network protocol of the attacked host to send repeated service requests repeatedly so that the attacked host cannot process the normal requests of users in time;Utilize the flaws of the attacked host’s Internet to repeatedly send malformed attack data, thus occupying most of the host’s memory and crashing the host.

The satellite network is characterized by decentralized users and a high degree of node autonomy, and the trust mechanism between nodes has become a major network security risk, especially for DDoS attacks, which have natural vulnerabilities. The study of DDoS attacks on LSCNs is still in its early stages at the moment. Similar to the terrestrial Internet, LSCNs are mainly subject to the following four popular DDoS attacks [20,25]:ICMP Flood: This attack sends a large number of ping packets to the victim in a short period of time and uses the method of exhausting the victim’s resources to achieve the purpose of paralyzing the server so that it cannot continue to work normally;TCP SYN Flood: This attack captures the defect of the TCP three-way handshake and four-way teardown protocol and initiates many false SYN connection request packets to the target host, which continuously occupies the resources of the target host and eventually causes the network to be paralyzed;UDP Flood: This attack sends a large number of UDP packets to the victim in a short period of time, making the victim overloaded and unable to undertake normal transmission work, exhausting the resources of the target host;HTTP Flood: This attack floods normal services in the network by sending malformed HTTP protocol packets, which can cause more damage without high rates.

## 3. Platform Design and Simulation

### 3.1. Simulation Tools

The LSCN security performance simulation platform is jointly implemented by STK, NS3, and Exata. Exata is a network simulation tool developed by Scalable Networks Technologies (SNT). Compared with QualNet, the performance of Exata has been further improved, with the advantages of parallel distributed simulation, real-time semi-physical simulation, and high-precision large-scale simulation. The packet-level simulation is carried out through NS3, and the importance of key satellite nodes in the region to the regional network is studied and analyzed. For LSCN network communication, DDoS attack deployment, and network-wide data analysis, Exata is utilized as a platform for network simulation. STK is used to generate the orbital information and data of the LEO and import them into Exata.

### 3.2. Orbit Parameter Settings

During the whole process, we set the satellite orbit inclination to 42° and the satellite orbit altitude to 335 km on STK. We select the network communication between A and B to implement an LSCN DDoS attack and analyze its security performance against DDoS attacks. After calculation, a total of 207 ground co-track satellites are selected to pass through the area within 10 s, and the simulation area is a rectangular area of about 8544 km × 4589 km. The orbit parameter information and 2D simulated scene are shown in Table 1 and Figure 3.

### 3.3. Interface Configuration

We use STK to establish a common-track satellite constellation that returns to the ground. Since the logical positions of the satellites in the constellation are relatively static within the time slice, as long as the relative positions of the satellites remain unchanged, the antenna can achieve communication as long as the antenna is kept aligned. The set antenna gain is 60 dB. We enable the QualNet interface in STK to configure the interface of the LEO satellite constellation and adopt the Open Shortest Path First routing protocol (OSPFv2) and distance−vector routing protocol (Bellman-Ford).

### 3.4. DDoS Attack Model Parameter Configuration

Since there is no Exata interface in STK that can be used directly, according to the use documents of the two simulation software programs, their joint simulation can realize interoperability by using the QualNet interface in STK and the AGI Satellite Toolkit Interface in Exata. The attack method chosen in this paper is a basic attack in Exata, and the data rate is set to 100 pkts/s. Layer transmission protocol (CBR) application between satellite No.207 and satellite No.66 is established. The application transmission start time is 1 s, the end time is 3600 s, the data of each packet is 2047 bits, the transmission interval is 1 s, and the routing strategy is the shortest-path strategy. According to the constructed space-time diagram key-node identification model, the key nodes of the entire satellite network are generated through the NetworkX and MATLAB tools, using node degree centrality and average betweenness centrality as indicators, and the importance of key nodes in the AB regional network is sorted. Finally, satellite No.67 and satellite No.119 are selected as the key nodes of the link, which are the targets of this simulation. On Exata, No.208, No.209, and No.210 node devices are set up to simulate a DDoS attack launched by the botnet on the No.67 satellite, while No.211, No.212, and No.213 are bots that launch attacks on the No.119 satellite.

## 4. Result and Discussion

In order to better verify the attack effect, this simulation attacked two satellite key nodes and multiple satellite key nodes under the OSFPv2 protocol and Bellman-Ford protocol, respectively, and analyzed the security performance of LSCNs after being attacked.

### 4.1. Two-Satellite-Key-Node Attack

Figure 4 shows the change in the transmission path of key satellite nodes No.119 and No.67 under the OSFPv2 protocol after the DDoS attack. The yellow * indicate the attacked satellite-key-node. The blue solid line represents the attack on the key node by the botnet host, the green solid line represents the original transmission path of the two satellites, the green dotted line represents the path change after being attacked, and the arrowheads represent the path direction (same as other Figures). Under normal transmission, the number of packets received by satellite No.66 is 3579. After attacking the key nodes of both satellites at the same time, the applied transmission link changes rapidly, while the number of packets received by satellite No.66 is 3483, which results in a low packet loss rate. As shown in Figure 5, under the Bellman-Ford protocol, after the attack, the number of packets received by satellite No.66 is 3539, which only has a certain impact. The average end-to-end delay under a two-satellite-key-node attack scenario is shown in Figure 6. It can be seen that only attacking two key satellite nodes of the whole local satellite network under the OSFPv2 protocol has little impact on its network security performance.

### 4.2. Multi-Satellite-Key-Node Attack

In order to verify the relationship between the number of key nodes and the attack effect, according to the ranking results of key nodes’ importance, the network security performance of several satellite key nodes under the OSFPv2 protocol after being attacked is analyzed.

As shown in Figure 7 and Figure 8, when DDoS attacks are applied to four key satellite nodes No. 121, No. 120, No. 119, and No. 67, the LSCN can still communicate by changing its satellite links. Although its communication links have changed, the number of satellite nodes that the path passes through remains the same, and so its communication propagation delay has no significant change.

As shown in Figure 9, Figure 10 and Figure 11, when the key nodes of satellites No.121, No.120, No.119, No.194, and No.67 were attacked, the number of nodes increased after the link path was changed. Figure 11 clearly shows that with the increase in the number of key nodes attacked, the propagation delay also increases gradually, from the initial 0.049 s to 0.087 s. In addition, the number of data packets decreases to 3519, and the attack effect becomes more and more obvious.

We continued to verify the relationship between the number of key nodes and the attack effect and continued to carry out DDoS attacks on 10 key nodes and 12 key nodes of the network. The findings demonstrate that when the number of key nodes of attacked satellites increases, the transmission delay between satellites increases, which is proportional to the number of attacked satellites. When the number of attacked satellites reaches 12, the latency rises to 0.1003 s. However, the packet loss rate hardly changes, which may be the reason why the satellite quickly switches the backup link after being attacked.

## 5. Conclusions

As a key strategic field in the information age, the LEO satellite constellation is facing increasingly serious network security problems. In this paper, the space-time diagram model is proposed to identify the key nodes in the LEO satellite constellation, and a DDoS attack is adopted as the main means to attack the key nodes. The attack effects of two-satellite key nodes and multi-satellite key nodes are verified, and the security performance of the whole satellite network against a DDoS attack is analyzed, which is of great significance to effectively deal with LSCN attacks.

## Figures and Tables

**Figure 1 sensors-22-07286-f001:**
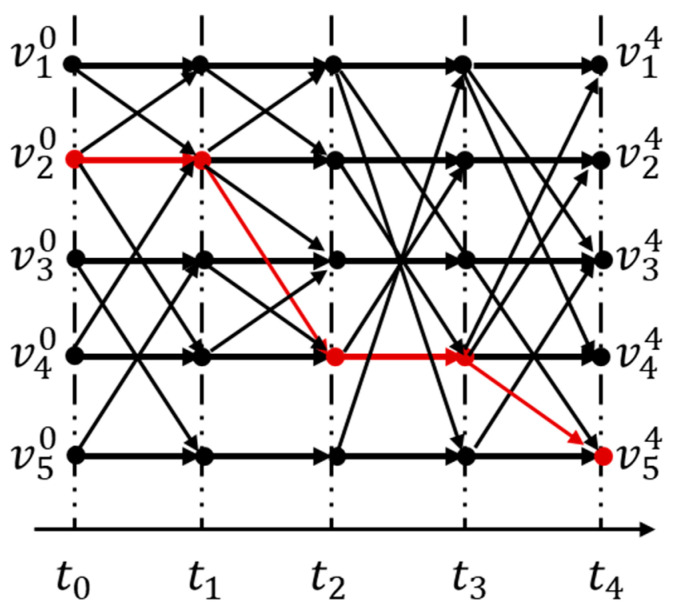
Space-time graph model.

**Figure 2 sensors-22-07286-f002:**
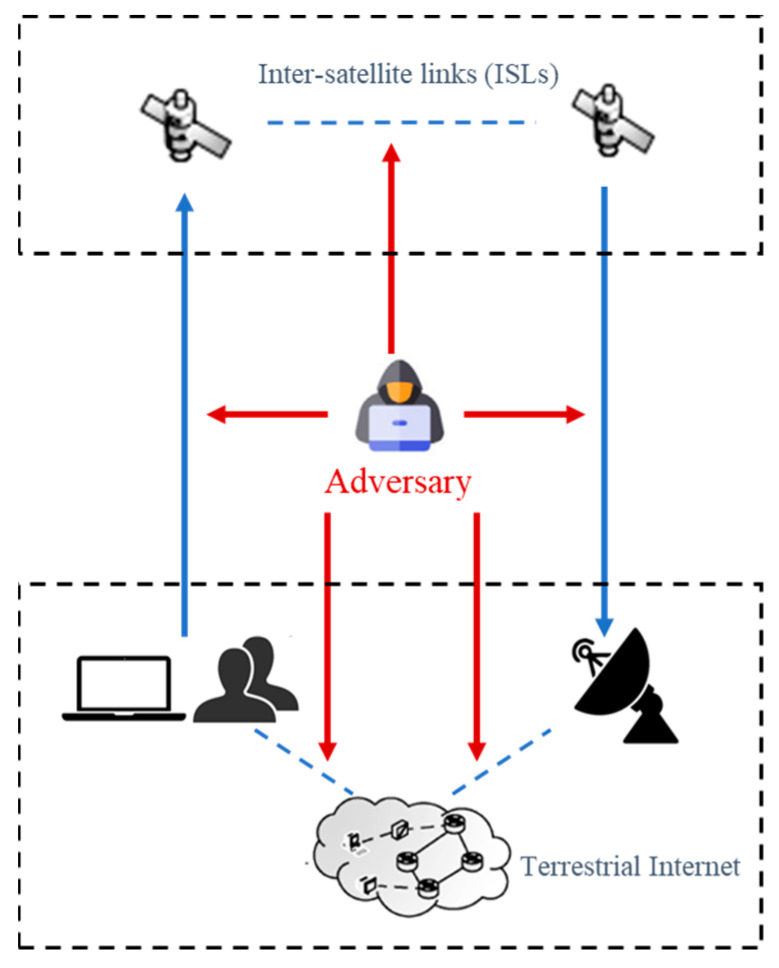
DDoS attacks on LSCNs.

**Figure 3 sensors-22-07286-f003:**
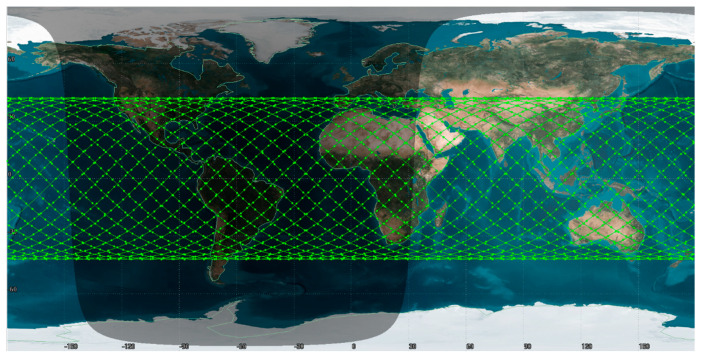
Simulated satellite 2D scene.

**Figure 4 sensors-22-07286-f004:**
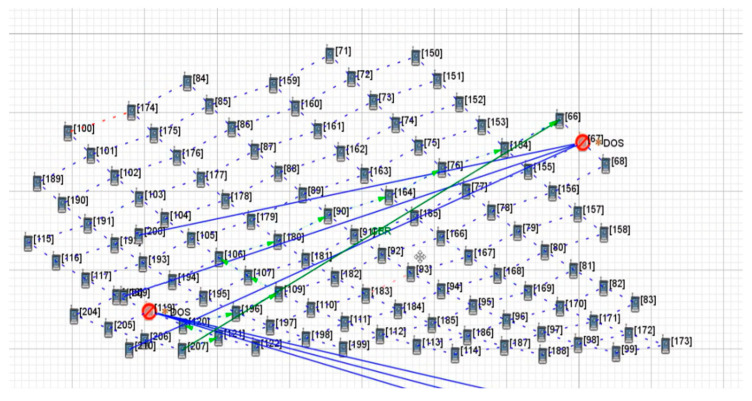
Two-satellite-key-node attack scenario under OSPFv2 protocol.

**Figure 5 sensors-22-07286-f005:**
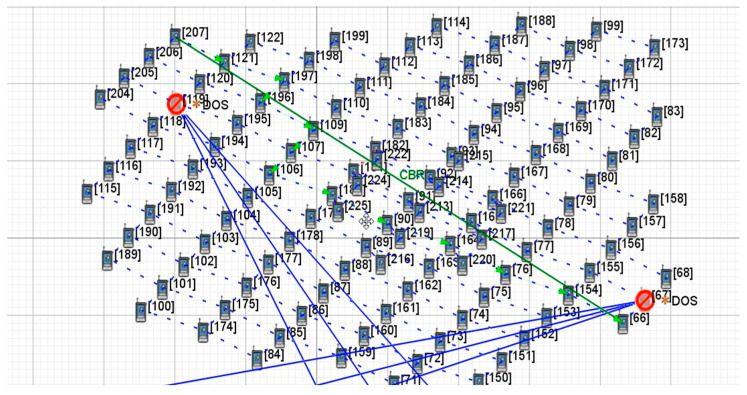
Two-satellite-key-node attack scenario under Bellman-Ford protocol.

**Figure 6 sensors-22-07286-f006:**
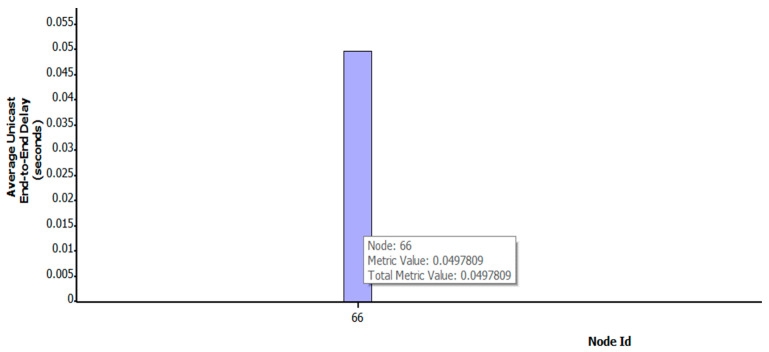
Average end-to-end delay under two-satellite-key-node attack scenario.

**Figure 7 sensors-22-07286-f007:**
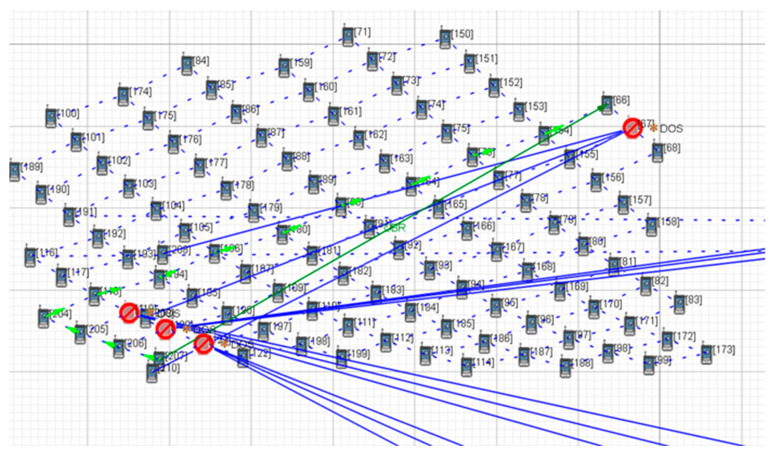
Four-satellite-key-node attack scenario.

**Figure 8 sensors-22-07286-f008:**
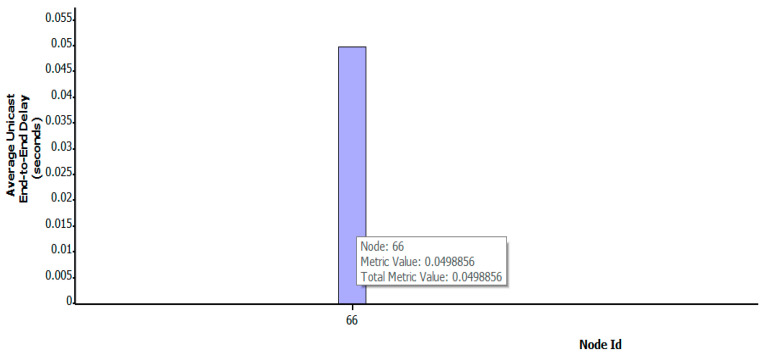
Average end-to-end delay under four-satellite-key-node attack scenario.

**Figure 9 sensors-22-07286-f009:**
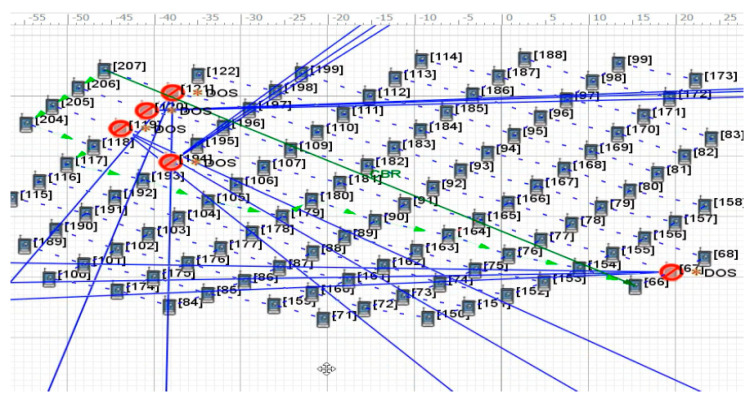
Five-satellite-key-node attack scenario.

**Figure 10 sensors-22-07286-f010:**
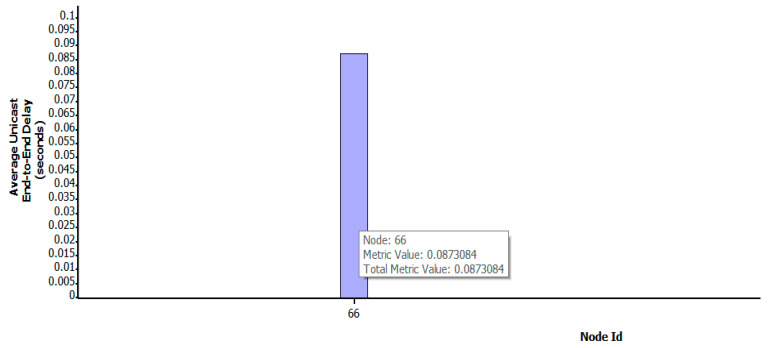
Average end-to-end delay under five-satellite-key-node attack scenario.

**Figure 11 sensors-22-07286-f011:**
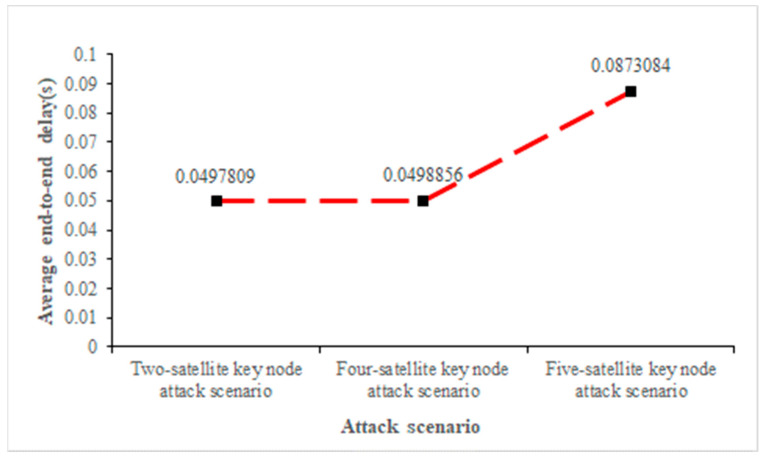
Average end-to-end delay under three attack scenarios.

**Table 1 sensors-22-07286-t001:** Orbit parameter information.

Parameter	Value
Orbit type	Circular
Altitude	335 km
Inclination	42.0 deg
RAAN	0 deg
Number of tracks	10
Number of satellites	207
Regional area	8544 × 4589 km^2^

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
