# Peer review of "Security Performance Analysis of LEO Satellite Constellation Networks under DDoS Attack"

_sensors, 2022, doi:10.3390/s22197286_

Round 1

Reviewer 1 Report

Security Performance Analysis of LEO Satellite Constellation

Networks under DDoS Attack

This paper looks at the simulation of possible attacks against LEO sattelite cliusters, considering the latter as a form of Adapticve Mesh metwork

While there is information 

The statement made (p1,ln41) is more related to traffic volummes. Any well designed network should haev a management  channel running OOB , or at the elast on the equivilent of a vlan, so it becomes a traffic capacity problem.  Almsot all of these can be dealt with with suitable firewalls on the devices, or even filteres pushed to groundstation nodes which prohibit direct communication to the active interfaces of a sat node - if one treats them as a traditional router or L3 switch - the principles for securing a management interface are similar.  A series of ports  and the backplane switch fabric can handle data rates orders of magnitude more than the tupical management interface. Consideration for queue strategies shoudl also be considered for protecting the interface,  RF spectrum I woudl imagine is appropriately reserved for OOB coms. It shoudl be near impossibel to run an attack against the sat units themselves. This leads to some concerns about the dos attacks chosen, its highly unlikly that a sat will be running a webserver ( hentce the HTTP flood is irrelivant). You need to make a much stronger case for the relevance of these attacks, as wella s provide far more detailed metrics as to bitrates, and clearly define the difference between volumetric attacks and those that cause resource stravation ( eg syn flood).  the DNS query flood is not relevant, and should be removed or fully motivated as to why its a viable test case.

Figures 4,5,7,9 there should be some further discussion as to how the reader shoud linterpret these images, and what the different lines mean, at least for the first one  you shoudl provide suitable discussion/walkthough of the image to enable a reader to get appropriate and neccessary value from later images.

Figures 6,8,10  should be combined to enable the reader direct comparison between the scenarios.  It is unclear is thisis OSPF or Bellman-Ford ( the final scenario makes no specification at all and the first two its unclear as to which the results relate to) At this point these really just take up space that could be used for expanding some of the issues above.

p1  

ln 29 - Consider these references carefully they do not support firectly the point being made. some variance of authors would be good.

ln 36 - citation to back up the claim of near fibre parity ?

p2 

STK is just reference with no explanation or reference, while NS3 is more wll known it similarly should be introduced. Exata, is again something that many readers will not be familiar with, it should be introduced, explained and cited. Specificlaly you shold mention that the cyber modules are laoded and how they are configured.

You need to be clar as to whay and how the porbital parameters presented were determined, and what imact these would have on the simulation. you need to assume theat the majority of readers do not have an background in orbital systems. similarly how was the gain determined ? what other values were tried, how would this imact the scenario

In the attack scenarios - mentionis made of specific noed, which even in the worsrt cas are <3% of nodes. How were these nodes chosen? how many times was the experiment run? what were the error marchign, spread of data ( a box/whisker plot would be ideal here). Why were these routing models chosen ( especially given that the internet at large uses BGPv4 ?

p4

"four-way wave protocol," this is possible better rephrased as 4-way teardown. but its not clear how the chosen attack has relevance wrt teardown.

P7

Which routing protocol do figures 7 and 9  represent ? Figure 10 ( and the relted ones) you mentaion an average , but where are the min/max, what was the distribution or ranges - a histogram with a single value is not a good way to present this data. How is the average calculated ?

Reviewer 2 Report

This paper proposes a space-time graph model for capturing the dynamics inherent to a LSCN under DDOS attack for the purposes of security evaluation. 

The paper's purpose and motivation are reasonably described but could be improved. The literature review could be improved. The experimental setup and process could also be improved.

Specific comments:

1) Space-time graph model is adopted for capturing LSCN dynamics. More discussion with reference to relevant existing literature is needed to explain why this model is selected above other possible models. First of all, other potential models should be enumerated and a compare/contrast discussion should be added to elucidate the advantages of the space-time graph model.

2) The space-time graph model has been previously proposed (referenced as articles #23,24). So it appears that this paper is simply using this model to simulate and analyze two artificially-constructed LSCN's and an artificially-constructed DDOS attack. Some discussion on the contributions that this paper adds to the field is warranted. Is the application of the STGM to the LSCN problem unique and significantly different than other applications STGM has been applied to? Have any additions to the STGM been made to accommodate LSCN and/or DDOS attacks been made? How would a reader benefit from the process and/or results of the study? If the purpose of the study is to develop a process/method for analyzing the security of LSCNs under DDOS attack, then this should be stated and the proposed process should be explicitly enumerated with enough detail so that it may be repeated by others.

3) While the authors mention that LSCN has few studies focusing on security performance, "few" implies that some exist. If these exist, they should be discussed. If no studies exist, then this should be stated.

4) The LSCN networks being analyzed and the DDOS attack simulated should be further justified. Are the simulated scenarios realistic, i.e., do they represent common LSCN and DDOS attack scenarios? If so, relevant literature references should be given. 

5) In the abstract, it is stated "Distributed denial of service (DDoS) attack has been considered as the most threatening attack method in the field of Internet security recently". This is a strong statement that must be defended. What about supply chain attacks? What about ransomware attacks? DDOS attacks are certainly relevant and are worth studying with respect to LSCNs, so I suggest that this statement is unnecessary and should be altered/removed. 

6) English grammar should be improved in several places in the paper. 

Round 2

Reviewer 1 Report

Security Performance Analysis of LEO Satellite Constellation 2

Networks under DDoS Attack

Revision 2

MDPI Sensors

https://susy.mdpi.com/

This paper has partially addressed changes noted by reviewers. There is still little clarity on the rationality behind the denial of service attacks against the sat nodes or any discussion of the need and difference between common data transmission paths and OOB management. I remain sceptical as to the specific attacks mentioned. For this kind of system, volumetric attacks would be likely the only concern (assuming management interfaces are appropriately isolated and secured.) Low bandwidth high-packet rate attacks such as syn floods could however cause issue with NAT (or hopefully CGNAT) state tables. There is no clarity of what attacks were applied in the three scenarios presented. what the data rates where, (both in bytes/s and pkts/s). AS such I can see this being near impossible for other researchers to replicate and arrive at similar results. What platform was the simulation run on.

In presenting the results, figures 6,8, and 10 should be removed, and either combined, or preferably replaced with a summary table clearly listing the relevant value along with the maximal data rates and the deviation in these from baseline conditions.   It is important to evaluate these against a 'normal' traffic load (how does performance look at 0,10,25,50, 75 and 85% of path capacity?).  How bad is the impact of the attacks at these various base traffic levels.  Given it is a simulation it is also important to disclose the specifications of the simulation platform. How is it ruled out that the simulation itself could be a bottleneck.

In terms of the path renegotiation, authors should be clear what triggers this - ie nodes becoming unresponsive or failing some capacity/latency threshold.

While there is a marked improvement, the work still leaves a number of questions unaddressed, and the final presentation of data needs to be considered.  The work should clearly recognise the limits and scope of the experimental work.

ln 15 - attacks have

ln 34 - LSCN not expanded on first use

ln 36 reference needed

ln 37  has its  change to have

ln 39 - what other references state there is a extreme vulnerability. the cited paper seems to refer to authentication not general threats.

ln 60 "Computer Systems International Top conference" this makes no sense and is incorrect. rather refer to the paper than the venue. As it stands, th "they" in the sentence following is not clear (referring to the conference in the previous sentence), and should be clearly referring to the authors of 17. strongly suggest rewriting these sentences.

ln 74 - "there are few research" to "there is little research". Security state rather than performance?

Figure 1 - expand the title to state clearly what is shown by the red path indicate, alternately consider expanding the sentences in 108-110 to explicitly state this.

ln 113 - This still fails to make clear that the attack being considered is on the data link channel and not the OOB management?

ln 121

line 122 - you are talking about server crash, but in effect the sat nodes are acting as switches, which typically have orders of magnitude higher rate processing, do to the lack of payload processing and asics for the packet inspection, switching and routing that must occur.

ln 135 - [16,21] rather

ln 139 and 146 - these are things that should have zero impact on the sat network as a transport layer, again with the only time any services being exposed should be on an OOB management interface. If that is being attacked the entire system has larger issues. If you are going to talk about this, clearly state how this is relevant to the sat transport as opposed to any generic network transport

ln 165 - no clear justification or statement in paper as to why these parameters are chosen. STK not expanded. Text is largely a restatement of table 1.

ln 167 "Select the network communication between A and B to implement LSCNs DDoS attack, and analyse its security performance against DDoS attacks" - this seems to have no place here? 

ln 188 "Establish an application layer transmission protocol (CBR) application between satellite No.207 and satellite No.66." rephrase as An application layer transmission protocol (CBR) was established satellite No.207 and satellite No.66.  Explain why these two were chosen. Does the choice of nodes matter?

fig 4 and 5 need to have much clearer titles and descriptions of what is going on. This needs to be linked to the text as well. Clearly indicate what symbols are used. Your readers should not be required to deduce this themselves. This is picked up in lines 212-214, but needs to be expanded. In particular the dotted green line is not clear at all and consists of directed arrow heads?

Figure 6 - this remains pointless as there is nothing to compare here. Why is a plot needed? I'm confused as to the need for the metric/nonmetric values shown in the label. what was the average unicast delay outside of the attack ie during normal operation . What about at 0, 50, 75 and 80% link loads ?

Fig 7 Renegotiated path seems to originate at sat 204 ( or its just not clear that the path is followed form 207 to 204. (fig 9 shown tis much better!)

A good metric would be to simply state is there a change in net path length on the recovered paths.

ln 222 is the word security needed here, there is nothing to do with security, just network performance.

Fig 8 - same applies as fig 6.

fig 10 - same as fig 6,8 

Refs 

[17] extra {} around ICARUS

Author Response

Please see the attached detailed response.

Reviewer 2 Report

The authors have addressed the comments sufficiently and have improved the paper.

Author Response

We sincerely appreciate your consideration for publication of our paper.